# Environmental Factors Impacting the Development of Toxic Cyanobacterial Proliferations in a Central Texas Reservoir

**DOI:** 10.3390/toxins16020091

**Published:** 2024-02-06

**Authors:** Katherine A. Perri, Brent J. Bellinger, Matt P. Ashworth, Schonna R. Manning

**Affiliations:** 1Department of Biological Sciences, Institute of Environment, Biscayne Bay Campus, Florida International University, 3000 NE 151st St., North Miami, FL 33181, USA; scmannin@fiu.edu; 2Watershed Protection Department, City of Austin, 505 Barton Springs Road, 11th Floor, Austin, TX 78704, USA; brent.bellinger@austintexas.gov; 3UTEX Culture Collection of Algae, College of Natural Sciences, University of Texas, Austin 204 W 24th Street, Austin, TX 78701, USA; mashworth@utexas.edu

**Keywords:** freshwater cyanobacteria, cyanotoxins, dihydroanatoxin-a, harmful algal proliferations, freshwater ecology

## Abstract

Cyanobacterial harmful algal proliferations (cyanoHAPs) are increasingly associated with dog and livestock deaths when benthic mats break free of their substrate and float to the surface. Fatalities have been linked to neurotoxicosis from anatoxins, potent alkaloids produced by certain genera of filamentous cyanobacteria. After numerous reports of dog illnesses and deaths at a popular recreation site on Lady Bird Lake, Austin, Texas in late summer 2019, water and floating mat samples were collected from several sites along the reservoir. Water quality parameters were measured and mat samples were maintained for algal isolation and DNA identification. Samples were also analyzed for cyanobacterial toxins using LC-MS. Dihydroanatoxin-a was detected in mat materials from two of the four sites (0.6–133 ng/g wet weight) while water samples remained toxin-free over the course of the sampling period; no other cyanobacterial toxins were detected. DNA sequencing analysis of cyanobacterial isolates yielded a total of 11 genera, including *Geitlerinema*, *Tyconema*, *Pseudanabaena*, and *Phormidium*/*Microcoleus*, taxa known to produce anatoxins, including dihydroanatoxin, among other cyanotoxins. Analyses indicate that low daily upriver dam discharge, higher TP and NO_3_ concentrations, and day of the year were the main parameters associated with the presence of toxic floating cyanobacterial mats.

## 1. Introduction

The global phenomena of cyanobacterial harmful algal blooms (cyanoHABs), occurring in the plankton, and cyanobacterial harmful algal proliferations (cyanoHAPs), arising from benthic mats, are increasing with anthropogenic eutrophication, climate change, and watershed development [1]. The impacts of cyanoHABs and cyanoHAPs include, but are not limited to, loss of surface waters for municipal and recreational purposes, chronic and acute health issues for humans and animals, and economic losses [2,3,4]. Globally, cyanoHAPs have been linked to rapid animal mortality after ingestion of cyanobacterial mat material and exposure to the potent neurotoxin anatoxin-a (ATX) [1,5,6]. The detection and reporting of the four ATX congeners (ATX, homoanatoxin-a, dihydroanatoxin-a, and dihydrohomoanatoxin-a) [7] has increased over the last decade, and reports have shown that the congeners are produced as mixtures in environmental samples [8,9,10].

Benthic and planktonic cyanobacterial genera producers of ATX congeners from the orders Nostocales and Oscillatorales include *Aphanizomenon*, *Cuspidothrix*, *Cylindrospermum*, *Dolichospermum*, *Geitlerinema*, *Hydrocoleum*, *Microcoleus*, *Microseria*, *Oscillatoria*, *Planktothrix*, *Phormidium*, *Pseudanabaena*, *Raphidiopsis*, and *Tychonema*, with *Tychonema, Phormidium/Microcoleus,* and *Oscillatoria* being the most frequently reported benthic genera [1,11]. Genetic identification of cyanobacterial taxa present in blooms or proliferations has typically utilized ribotyping via 16S, 23S, and/or ITS sequence-based analysis, and metagenomics of diverse communities has grown in popularity [12,13]. While these tools are useful for identification of taxa present in a bloom or proliferation, isolating cells and filaments into culture is needed for the further characterization of individual cyanoHAB/HAP species, including growth, genetic, and biochemical analyses.

Environmental drivers of freshwater cyanoHAB/HAPs include anthropogenic eutrophication, low flows or flushing rates through inhabited systems, and reduced turbulence and water column stability, factors being exacerbated by climate change and non-native species invasions [14,15,16]. Nutrients, i.e., nitrogen (N) and phosphorus (P), are recognized as limiting algal production, but anthropogenic eutrophication afflicting more water bodies via point (e.g., wastewater discharges) and nonpoint (e.g., agricultural runoff) sources is likely to be the primary cause of toxic cyanobacterial events [17]. Changes in rainfall patterns from smaller long-duration to larger short-duration events following dry periods can result in larger floods which mobilize increased amounts of sediment downstream, increasing the loading of nutrients into receiving water bodies [18]. Given the increased risk of toxic cyanoHAB/HAP events in water bodies utilized for a variety of ecosystem services, there is a need to be able to identify drivers associated with increased cyanobacterial biomass, taxon composition, and toxin production.

Toxic benthic cyanobacterial mats were linked to numerous dog deaths in a two-month period of 2002 in the Tarn River, France [19], and animal deaths in two New Zealand rivers and farming stock ponds in the late 1900s and early 2000s [9,20,21]. More recently, dog deaths were reported in Berlin, Germany in 2017 [22] and Austin, Texas in 2021 [23]. The present work provides an expanded report of initial findings regarding a HAP event in Lady Bird Lake, Texas that resulted in several canine deaths [24], including an analysis of environmental factors and the presence of dihydroanatoxin-a (dhATX).

The Lady Bird Lake reservoir flowing through the heart of Austin, Texas, USA features a 10-mile hike-and-bike trail and multiple parks (Figure 1). The trails and parks are especially popular with dog owners and dogs frequently wade into the nearshore waters. The water quality of the reservoir meets State of Texas contact recreation standards for *E. coli*, but recreational swimming has been banned in the reservoir since 1964 because of risks posed by high flows and bottom debris [25].

In 2019, five dog deaths and four dog illnesses were reported in July and early August, mostly from one weekend, at two popular parks where reservoir access for dogs is allowed. Observations of thick cyanobacterial mats were documented (Figure 2) and subsequent testing detected dhATX in mat samples [24]. After the reports, one park was completely closed, other dog access points were closed, and warning signage was placed around the reservoir.

This study investigated environmental and nutrient conditions associated with the 2019 cyanoHAP event of Lady Bird Lake [24]. Observations were contrasted with historical data (2016–2018) to identify factors that may have contributed to the formation and accumulation of metaphyton materials. Due to the presence of mats in still waters in the mid-summer, it was hypothesized that low-flow, warm water conditions with sufficient nutrients would promote mat formation. Thus, cooler temperatures and increased flows would result in fewer mats. Environmental conditions measured included water nutrient chemistry and average daily discharge rates from the upriver dam. Given the lack of reported dog illnesses or deaths along Lady Bird Lake in previous years, it is possible that little to no toxic cyanobacterial mat material was present prior to 2019 and/or environmental conditions did not favor the accumulation of metaphyton materials. Two potentially important antecedent events prior to 2019 included an infestation by zebra mussels, known ecosystem engineers, and a record flood that contributed significant sediment to the reservoir. Cyanobacterial isolates present in the metaphyton mats were identified using molecular methods and a DNA fingerprint library of the primary species of concern from the 2019 cyanoHAP event was constructed. Due to the paucity of environmental studies associated with cyanoHAPs globally, this information will contribute to better understanding the emergence, duration, and potency of these potentially harmful events.

## 2. Results

### 2.1. Species Information

The mats were complex assemblages, i.e., metaphyton, consisting of bacteria; unicellular, filamentous, and colonial cyanobacteria; filamentous green algae; and diatoms (Figure 3).

DNA sequenced from strains isolated from the 2019 event suggested at least six cyanobacterial operational taxonomic units (“OTUs”) present, based on sequence similarity (Figure 4A). A consensus taxonomy of the OTUs based on the top similarity scores was generated by the National Institute of Health’s Basic Local Alignment Search Tool (BLAST) to identify two of these as representatives of the *Geitlerinema* (Anagnostidis & Komárek) Anagnostidis and *Pseudanabaena* Lauterborn genera. The remaining OTUs include a leptolyngbyacean, one pseudoanabaenacean, one oscillatorian, and one OTU with genetic similarity to pseudoanabaecean and wilmottiacean genera [26].

Among the reisolates from the strains isolated in the 2019 event, five OTUs were identified based on 16S ribosomal DNA sequence similarity (Figure 4B). The consensus taxonomy of these OTUs based on the top BLAST scores suggested an identity of these OTUs within the *Merismopedia* Meyen, *Leptolyngbya* Anagnostidis & Komárek, and *Pseudanabaena* and *Synechocystis* Sauvageau genera, with an additional leptolyngbyacean OTU. The 23S ribosomal DNA results were not considered for OTU assignment, as BLAST scores tended to be significantly lower than those of the 16S data strains [26].

### 2.2. CyanoHAP Toxicity

Surface coverage of metaphyton was inconstant at each site, dependent on wind and flow conditions. The highest contents of dhATX (>133 ng/g wet weight [WW]) were observed from mat materials collected at sites 1996 and 1252 between August 6th and 12th, 2019. After a brief absence of mats from all sites, the presence of dhATX (0.6–19 ng/g WW) was again reported at sites 1996 and 1252 between September 30th and October 14th. No metaphyton was observed after October 14th and water samples were negative for toxins throughout the observation period. No toxins were detected in mat samples collected from sites 1671 or 1997 throughout the monitoring period.

### 2.3. Flow Conditions

In 2019, reports of dog illnesses and subsequent collection of mat materials were associated with daily average discharges from the Tom Miller Dam dropping below 10 m^3^/s in early July (Figure 5A). The 2019 discharges associated with cyanoHAP presence were more than 10× lower than during the same time frame in 2016–2018 (Figure 5B). Decreased mat presence late in the summer and in the fall were associated with increased discharge (>10 m^3^/s) from the Tom Miller Dam (Figure 5A).

### 2.4. Water Quality Conditions

At the sites sampled on the weekend of the first reported dog death, water temperatures exceeded 30 °C. Over the course of the monitoring period, average water temperatures did not differ in 2019 ($\bar{x}$ = 26.6 ± 2.5 °C) from the previous three years ($\bar{x}$ = 25.8 ± 1.7 °C) (*t_crit_* = −1.038; *p* = 0.31).

Among cyanoHAP sites, 1996, where the most dog deaths occurred, had the highest ammonia nitrogen (NH_3_-N), total Khjeldhal nitrogen (TKN), and total phosphorus (TP) concentrations and the lowest molar nitrogen-to-phosphorus (N:P) ratios, nitrate+nitrite nitrogen (NO_x_-N), and total nitrogen (TN) concentrations (Table 1). The water chemistry was similar between sites 1252 and 1997, though cyanoHAPs were only observed at the former site (Table 1). Site 1671 had the highest NO_x_-N and TN concentrations and highest N:P ratios. Soluble reactive phosphorus (SRP) was below detection limits at all long-term and event sample sites (<0.08 mg/L), and the average molar N:P ratio was indicative of P-limitation across all sites.

Among the Austin Lake Index (ALI) sites, total suspended solids (TSS; *t*_crit_ = 2.3; *p* < 0.05), NH_3_ (M-W U = 57; *p* < 0.05), and NO_x_ concentrations (M-W U = 39; *p* < 0.01) were significantly higher, and molar N:P ratios were significantly lower (M-W U = 48; *p* < 0.05; Table 1) in 2019 than in the 2016–2018 period. Total P concentrations in 2019 were marginally higher relative to the previous three-year period (M-W U = 61; *p* = 0.05).

### 2.5. Water Quality Ordination

Environmental parameters collected at ALI and cyanoHAP sites resulted in distinctive groups in ordination space driven by nutrient concentrations, day of the year, and average daily discharges (Figure 6). In general, ALI sites prior to 2019 experienced higher flows and lower TP and NH_3_ concentrations relative to the summer 2019 ALI and the cyanoHAP samples collected from sites 1996 and 1252. The cyanoHAP monitoring sites that were negative for toxins in 2019 (1671 and 1997) were influenced by higher NO_x_, TN, and N:P ratios (Figure 6) [26].

## 3. Discussion

The 2019 cyanoHAP event in Lady Bird Lake was the first documented toxic cyanobacterial event and incidence of dog deaths in the basin and occurred with cohesive cyanobacterial mats that developed on the benthos and floated to the surface. During a previous record drought between 2009 and 2015, planktonic cyanobacterial biomass exceeded World Health Organization thresholds of concern, but the blooms remained negative for toxins [27]. It was hypothesized that environmental conditions, notably nutrients, were not adequate for the blooms to produce toxins, though the cyanobacterial assemblage may not have contained toxigenic species at that time. Throughout 2019, surface waters remained negative for toxins and the reservoir was open for other recreational uses (e.g., kayaking and stand-up paddle boarding), but uncertainty about the full extent of the cyanoHAP event and possible exposure to toxins resulted in a decline in recreational use determined through rental of watercraft [28,29]. Parks were not fully opened until the end of November 2019.

From the cohesive mats, dhATX was the only toxin detected; there were no other ATX congeners present. Detection and prominence of dhATX appears common in environmental mat samples [6,30] but has not previously been reported as the sole congener detected. Dihydroanatoxin-a had previously been considered a less-lethal break-down product of ATX [31]. However, recent research shows that dhATX is an actively produced metabolite [32] and may be 4× more toxic by oral ingestion than ATX [11]. Significant amounts of dhATX were measured in mats from farm ponds where multiple dog deaths occurred in New Zealand [9,33,34]. The large variance in dhATX concentrations over the observation period in this study has been observed between individual species in culture [33] and within mats between sampling systems [30].

Of the five main cyanobacterial genera present in the metaphyton mats, only *Geitlerinema* and *Microcoleus* (previously known as *Phormidium*) have species known to produce ATX and dhATX [1,11]. *Limnothrix* and *Pseudanabaena* had been isolated from planktonic water samples collected in upriver reservoirs within the Highland Lakes system from 2010 to 2015 [35], demonstrating that these taxa were present upriver of the current study site.

While new taxa were isolated over the course of the study, revealing cryptogamic cyanoHAP genera, there were many strains that grew poorly or were lost completely in culture. This demonstrates the difficulty of isolating and maintaining wild taxa for additional analysis and characterization. Thus, only the strains that were sustained in culture are reported for this study, and there are many diverse candidates not represented in the present DNA sequence data. For example, numerous filaments of the known dhATX-producer *Oscillatoria* were detected in micrographic observations (Figure 2). However, isolates did not perform well in culture; thus, there are no DNA sequence data representing these taxa. To overcome challenges of isolation and maintaining cultures, future monitoring will employ metagenomics to analyze environmental DNA to better capture the complex cyanobacterial consortia that comprise these metaphyton mats.

Toxic cyanobacterial mats and cyanoHAPs remain understudied relative to planktonic blooms globally. However, increasing occurrences in New Zealand, Europe, and now in the United States (e.g., Zion National Park and additional reservoirs in central Texas) are prompting more research into possible drivers of these unique growth forms. Based on initial research, it appears that stressors commonly associated with planktonic cyanoHABs are also likely to be important in cyanoHAP ecology [36]. For example, rapid flows will scour sediments and flush and disintegrate floating mats. With elevated flows early in the summer of 2019, there were no reported dog exposures, presumably due to an absence of abundant mats nearshore. It was not until flows declined in early July that enough mat material was able to develop and remain nearshore through the next month, enabling interactions with dogs entering and exiting the water. Relative to the previous 3-year period, flows in July 2019 were considerably lower (<10 m^3^/s). In mid- to late-August, average discharge rates increased and were comparable to the previous years, and there was an absence of mat material for collection. The site with the largest extent of metaphyton biomass and where most dogs died, Red Bud Isle, is immediately below the Tom Miller Dam, making that area the most responsive to discharge volumes. With a decline in flow rates again in September, cyanobacterial mats were able to regrow, though toxin contents were generally lower, possibly attributed to cooling water temperatures or shorter daylight periods.

Nutrients are typically regarded as the most important driver of cyanoHAB/HAP events [36]. Though we did not see many significant changes in nutrient concentrations at the long-term monitoring sites in 2019 relative to the previous three years, we believe there are several factors obfuscating our ability to demonstrate meaningful changes in surface water quality (e.g., limited number of summer samples), and this study does not have any sediment nutrient chemistry. Mats at the sediment–water interface will experience nutrient availability in fundamentally different ways from planktonic species. Benthic mat species have the potential to access nutrients liberated from anoxic sediments, in the overlying water column, and from particulate inorganic and organic matter within the mat with extracellular enzymes [37,38]. The NMDS ordination analysis, though, showed that 2019 ALI and toxic cyanoHAP monitoring sites were influenced by elevated TP and NH_3_ concentrations, and lower average daily discharges late in the season, suggesting that two events may have contributed to the changes in water chemistry relative to previous years.

First, a significant flooding event in October of 2018 led to months of extreme turbidity as sediments from the basin were flushed through and deposited into Lady Bird Lake. Due to unsafe flow conditions, water sampling was not feasible, so there are no data for the sediment and nutrient load that occurred. Agriculture in the watershed is low relative to rangeland and is not considered the primary source of eutrophication as is the case in most watersheds globally. However, the sediments of the Colorado River basin are highly erosive and during large rain events a significant amount of sediment material is mobilized. Newly deposited particulate matter could provide a significant source of nutrients to fuel benthic mat development. Further, the influx of sediments in the fall may also transport and deposit cells/spores of HAB species that have been identified from the upriver reservoirs [34]. This dynamic highlights the importance of monitoring precipitation events, flooding, the frequency and volume associated with discharge events, and the potential for sediment deposition that could fuel benthic growth. These variables, along with nutrient inputs, are all contributing factors to the formation of cyanoHAPs and more measurements are needed to develop better models to help researchers understand these global phenomena [9,39].

Next, the establishment of zebra mussels in Lady Bird Lake around 2017 is the likely driver behind the significantly reduced TSS concentrations observed in 2019. Along with benthification of aquatic ecosystems, zebra mussels are proficient in excreting bioavailable NH_3_ and P after filtering phytoplankton from the water column [40,41,42]. Increases in nutrients at the sediment–water interface coupled with anaerobic microbial activity in nutrient-rich surface sediments would provide ideal growth conditions for benthic cyanoHAPs. However, the full extent of the ecosystem changes would likely be underestimated by the surface water sampling carried out in this study. Future research should target surface sediment nutrient chemistry to better understand the bioavailability of reduced nutrients to benthic cyanobacteria.

The near-daily hypolimnetic releases from the Tom Miller Dam, even if small, are an important source of nutrient loading, especially to Red Bud Isle, as there are no other tributaries near that site [27]. Ammonia tends to be scarce in freshwater systems and can be highly toxic itself, but cyanobacteria can efficiently and rapidly utilize NH_3_ [43]. Oxidation of NH_3_ to NO_3_^−^ would also be expected in the open waters. Therefore, NH_3_ loading in hypolimnetic waters coupled with zebra mussel excretion could explain the increased NO_x_ concentrations observed in 2019 relative to previous years. Effective use of NO_3_ by *Microcoleus* spp. (cf. *M. autumnalus* and *M. favosus*) via gene expression has been shown [38]. In culture experiments with *M. autumnalus* (previously known as *P. autumnale*), higher concentrations of dhATX over time were positively correlated with low-to-moderate concentrations of NO_3_ and orthophosphate (PO_4_), and dhATX concentrations subsequently declined at high NO_3_ concentrations [33]. The highest ATX values were observed after NO_3_ and PO_4_ concentrations had significantly declined [33]. Concentrations of NO_x_ in Lady Bird Lake in 2019 were elevated in comparison to previous years, but were 1–5× lower than reports in the literature [33,38]. This, combined with SRP concentrations that were below laboratory detection limits (i.e., 0.08 mg/L), suggests favorable conditions for increased dhATX production. Additionally, as Austin’s human and animal population has grown, increased localized nutrient loading from pet waste and lawn fertilizers could have also contributed to the proliferations [44].

The occurrence of a cyanoHAP event in one of the most popular reservoirs in the heart of the City of Austin, TX, USA has raised concerns that proliferations could re-occur annually, and impact future reservoir uses. The basin in Central Texas is known as “flash-flood alley” because of the interruption of long periods of low to no rainfall followed by large, punctuated rain events; and there are concerns that events like the one that occurred in 2019 could happen more often due to climate change. In addition, quagga mussels could infest the reservoirs, colonizing a greater area of the benthos, further exacerbating nutrient dynamics, possibly to the benefit of benthic cyanobacterial mats. These factors have prompted development of an annual monitoring program to track cyanoHAP and environmental patterns during the growing season to better model emergence, persistence, and cyanoHAP intensity. Additionally, P-mitigation measures using lanthanum-modified bentonite are being implemented to sequester sediment P pools in the hopes of eliminating an important nutrient reservoir for benthic mats.

## 4. Materials and Methods

### 4.1. Site Description

Lady Bird Lake, Austin, Texas, is the last in-line reservoir on the Colorado River (Figure 1). The reservoir is technically not considered a part of the Highland Lakes chain as it is overseen by the City of Austin, whereas the other impoundments were created by and are under the management jurisdiction of the Lower Colorado River Authority (LCRA). Inflows into Lady Bird Lake, however, come from Lake Austin through the Tom Miller Dam, which is regulated by the LCRA to ensure minimum environmental flow and downriver customer needs are met. The upriver hypolimnetic water chemistry, discharge regime, and broader basin environmental characteristics impact Lady Bird Lake. Barton Creek, a large tributary to Lady Bird Lake, has a watershed that has largely been left undeveloped through purchase of conservation lands (Figure 1). Conversely, two other large tributaries to Lady Bird Lake, Shoal and Waller Creeks, drain dense urban watersheds.

### 4.2. Sample Collection

Four sites (1996, 1671, 1252, and 1997; Figure 1) were sampled for cyanoHAPs on August 6th and 12th, September 9th and 24th, and October 2nd and 14th, 2019, after which no mats were observed. All sites chosen are popular recreational areas and included the two parks where the initial dog deaths/illnesses were reported (i.e., 1996 and 1252). Sites were accessed from the shore and mats were collected by scraping cohesive material off the surface sediments or rocks, or grabbing floating mats (i.e., metaphyton) and placing them into 250 mL amber plastic bottles for determination of cyanotoxins in the mats only. A separate 250 mL mat-free water sample was collected into a 250 mL amber plastic bottle for analysis of cyanotoxins in the water. Mat biomass was collected from the nearshore area, in 1 m or less of water, which would represent the material most likely to be encountered by a swimming canine or human.

Ambient water temperature (°C) was measured with a Hydrolab MS5 Datasonde (OTT Hydromet, Loveland, CO, USA). Water samples were simultaneously collected for nutrient analyses, except for the August 6th event, by taking surface (top 0.1 m) grabs in un-preserved and sulfuric acid-preserved 250 mL clear plastic bottles. Samples were kept on ice, and mat and water samples were delivered on the day of collection for cyanotoxin analyses. Water samples were also delivered to the LCRA Environmental Services Laboratory (ESL) on the day of collection and were analyzed for ammonia (NH_3_), nitrate/nitrite (NO_x_), total Khjedhal nitrogen (TKN), soluble reactive phosphorus (SRP), and total phosphorus (TP) concentrations (μg/L) by standard methods. Total nitrogen (TN) was determined as the sum of TKN and NO_x_.

### 4.3. Long-Term Monitoring

The Watershed Protection Department has carried out approximately bi-monthly water quality monitoring of Lady Bird Lake as part of a condition determination under the Austin Lake Index (ALI) program at three sites, two of which (#2 and #5) were utilized for comparison with 2019 data given the proximity of the sites to HAP monitoring locations (Figure 1) [45]. Surface (0.3 m) grab samples from near the center of the reservoir were collected, preserved in the field with concentrated H_2_SO_4_, and kept on ice and delivered on the same day as collection to the LCRA ESL for nutrient analyses. For the purposes of this study, mean nutrient concentrations for the period of 2016–2018 were compared relative to 2019 for the peak growing season months of June through October. The short duration of the “pre-cyanoHAB” period was caused by a record drought between 2009 and 2015, where discharges through the Highland Lakes were abnormally low to preserve water volume [26]. In response to the Highland Lakes being refilled in 2015 as a result of multiple large rain events, discharges through the reservoirs returned to “normal” (i.e., water was provided to downstream rice farmers in the spring and fall). This period also represents the initialcolonization and infestation by zebra mussels, with the reservoir being declared fully infested in 2018. Daily discharge volumes from the Tom Miller Dam were provided by the LCRA.

### 4.4. HPLC-MS Analysis

Biochemical analyses were performed to determine the presence/absence of targeted cyanotoxins (e.g., anatoxins, microcystins, cylindrospermopsin, and saxitoxins) in water and biomass samples, as previously published [23]. Data were analyzed using Agilent’s MassHunter Qualitative Analysis software (version 10.0). Raw and processed mass spectrometry data can be found in the corresponding dataset [26].

### 4.5. 16S and 23S rRNA Barcode Analysis

Individual cells and filaments were isolated from mat samples using a drawn-out glass Pasteur pipet, and isolates were maintained in 15 mL conical tubes containing BG-11 medium (utex.org). Subsamples of isolates were transferred into pre-labeled 2 mL screw-top centrifuge vials for DNA extraction and analysis. A second round of isolations was performed from existing isolates in 2021. To extract genomic DNA, lysis buffer (200 µL; 1 M NaCl, 70 mM Tris, 30 mM Na_2_EDTA, pH = 8.6) was added to the samples, which were then vortexed and centrifuged for 1 min at 16,500× *g*. Following centrifugation, the supernatants were removed and another 200 µL of lysis buffer was added along with acid-washed glass beads (400–600 µm, Fisher), 200 µL 24:1 chloroform:isoamyl alcohol, and 25 µL 10% dodecyltrimethylammonium bromide (*w*/*v* in double-distilled H_2_O). Samples were then placed in a bead beater for 40 s. Aqueous and organic phases were partitioned by centrifugation for 2 min at 2100× *g*. After partitioning was achieved, 100 µL of the top (aqueous) layer was removed and placed in a new 1.5 mL snap-cap centrifuge tube. The samples were then processed using a Gene Clean Turbo (MP Biomedicals, Tokyo, Japan) DNA extraction kit per the manufacturer’s instructions. Extracted DNA was quantified using 1 µL of purified DNA on a Qubit^TM^ 4 fluorometer using the Qubit^TM^ dsDNA high-sensitivity (HS) assay (ThermoFisher Scientific, Waltham, MA, USA). 

The 16S region was amplified by PCR for the 2019 isolates using a (GC)16Scya primer [46], and the 2021 isolates used a 16Scya primer [46]; 23S amplicons [47] were generated only for the 2021 isolates. PCR samples containing a single band were purified using a GeneJET PCR kit per the manufacturer’s instructions (ThermoScientific, USA). Purified samples were submitted for Sanger sequencing to the University of Texas-Austin DNA Sequencing Facility. Sequences were trimmed and assembled using Geneious Prime v2023.2.1. Assembled sequences were then aligned and a Neighbor-Joining (NJ) tree with 100 pseudoreplicates was constructed using the Jukes–Cantor model, also in Geneious Prime v2023.2.1. OTUs were determined from the NJ tree out of clades with 100% bootstrap support. A consensus taxonomy for these OTUs was determined from the most-conserved taxonomic level of the top BLAST scores for each assembled sequence [25]. Raw and trimmed sequence data can be found in the corresponding dataset [26].

### 4.6. Statistical Analysis

A student’s *T*-test was applied to compare ALI water quality data for the periods 2016–2018 and 2019. Data were log_10_(n + 1) transformed as needed to meet normality, but where normality could not be achieved, a Mann–Whitney (M–W) U test was used (α = 0.05) Statistical analyses were carried out with SigmaPlot v.14 (Systate Software, Inc., San Jose, CA, USA). Non-metric multidimensional scaling (NMDS) with a Bray–Curtis matrix was used to understand similarities and differences in environmental characteristics between the ALI and cyanoHAP monitoring sites for the period of 2016–2019. This data visualization provides a means of assessing similar and dissimilar characteristics of sites. The final dimensional analysis followed a preliminary analysis stepping down from six- to one-dimensional solutions composed of 250 iterations and 50 runs with real and randomized data. Sites were graphed in two-dimensional ordination space and vectors represent environmental metrics. The NMDS analysis was carried out with PC-ORD v. 6 (MJM Software Design, Gleneden Beach, OR, USA).

## Figures and Tables

**Figure 1 toxins-16-00091-f001:**
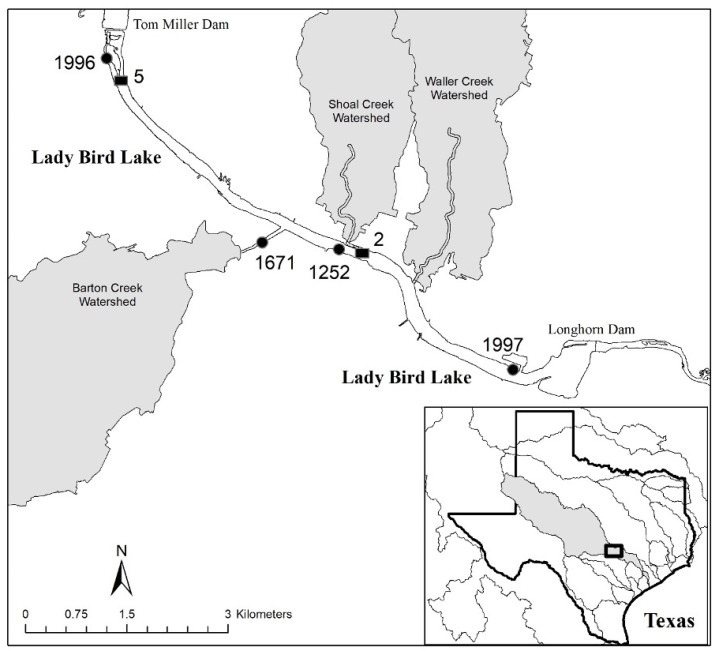
Long-term water quality monitoring sites (2 and 5) and 2019 cyanoHAP event sites (1996, 1671, 1252, and 1997) in Lady Bird Lake, and the location of that reservoir within the Colorado River Basin (inset). The three largest urban watersheds (Barton, Shoal, and Waller Creeks) flowing into Lady Bird Lake are also shown.

**Figure 2 toxins-16-00091-f002:**
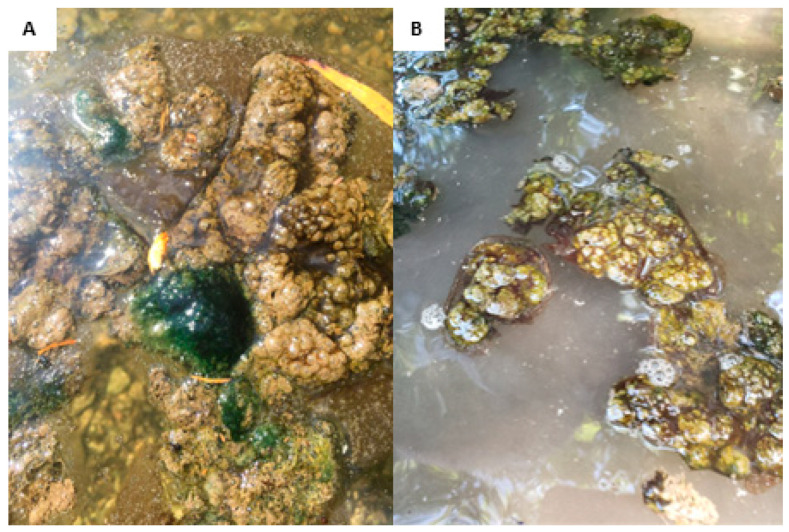
Photographs of metaphyton mats observed around Red Bud Isle, an off-leash dog park located on Lady Bird Lake near Tom Miller Dam in Austin, Texas, USA. Photographs were taken on August 8th (**A**) and August 12th (**B**), 2019.

**Figure 3 toxins-16-00091-f003:**
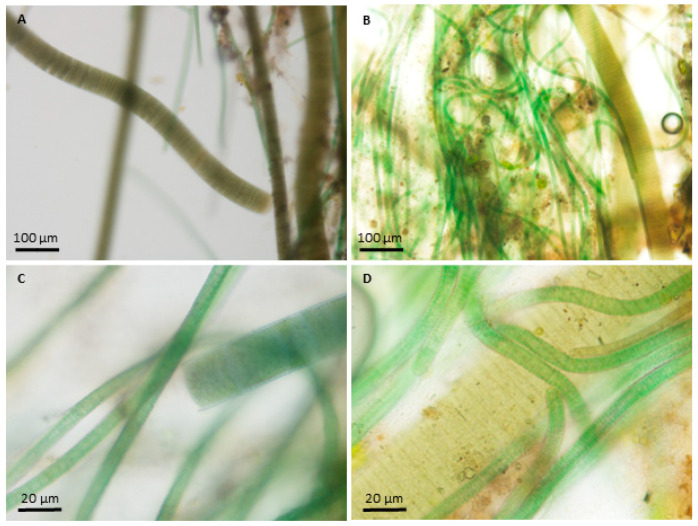
Micrographs of filamentous cyanobacteria present in the metaphyton mat materials collected from Red Bud Isle, Austin, Texas. Images were taken at 100× (**A**,**B**) and 400× (**C**,**D**). Metaphyton mats were composed of a complex community including numerous bacteria, diatoms, and abundant filamentous cyanobacteria. Oscillatoroid species were common; these included very large, broad olive-green filaments (e.g., *Oscillatoria*) as well as thin blue–green trichomes (e.g., *Microcoleus*). Many of these taxa were successfully isolated and cultured for molecular analysis.

**Figure 4 toxins-16-00091-f004:**
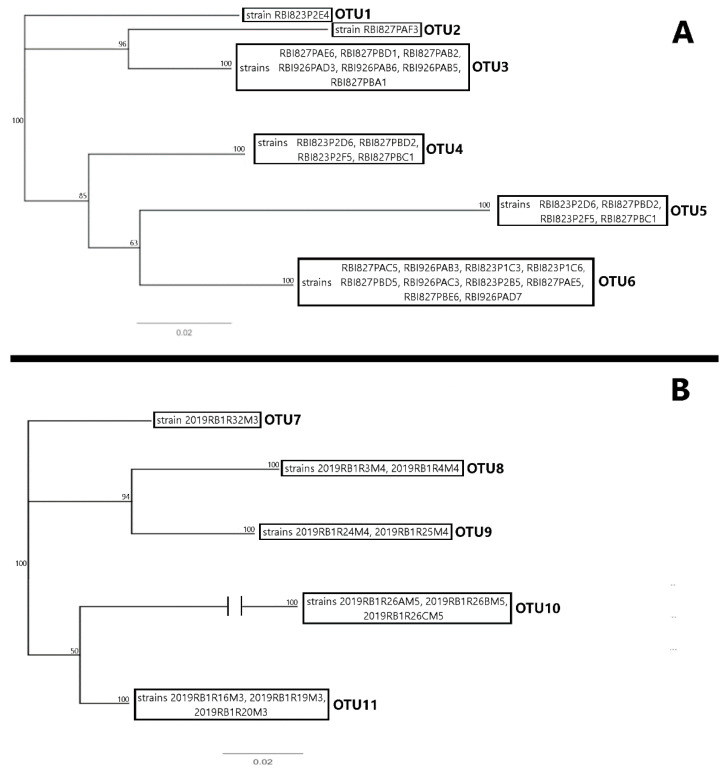
Distance trees calculated from 16S ribosomal small subunit sequence data collected from the cyanobacterial strains isolated from the bloom event. Panel (**A**) represents sequences from the initial 2019 strains. Panel (**B**) represents sequences from reisolations of the 2019 strains after the pandemic. Boxed terminal branches represent strains with identical or near-identical sequence data, which have been grouped into OTUs. Numbers at nodes represent bootstrap support for the indicated clades. Break in the OTU10 branch is for clarity; unmodified trees available in the supporting material provided for this study [26].

**Figure 5 toxins-16-00091-f005:**
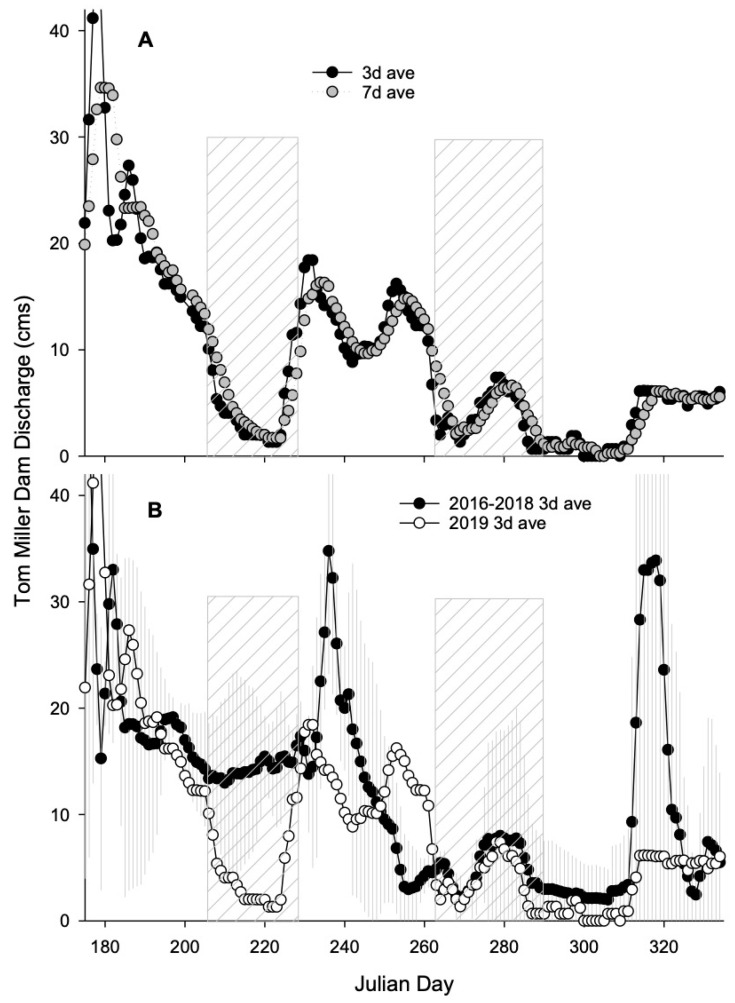
Discharge rates (cms, m^3^/s) are reported for Tom Miller Dam, Austin, Texas. Tom Miller Dam is a bottom-release dam directly upstream to Red Bud Isle, a park on Lady Bird Lake managed by the City of Austin that permits dogs to recreate off-leash and the primary location of exposure to metaphyton materials responsible for numerous dog illnesses/deaths. Panel (**A**) shows the 3-day (closed circles) and 7-day (gray circles) average (ave) discharges for 2019. Panel (**B**) shows the 3-day average discharges for 2016–2018 (open circles) and 2019 (closed circles). Diagonal boxes represent the period in 2019 when dog deaths/illnesses or cyanotoxins were directly measured, corresponding with lower-than-average discharge rates by comparison with previous years.

**Figure 6 toxins-16-00091-f006:**
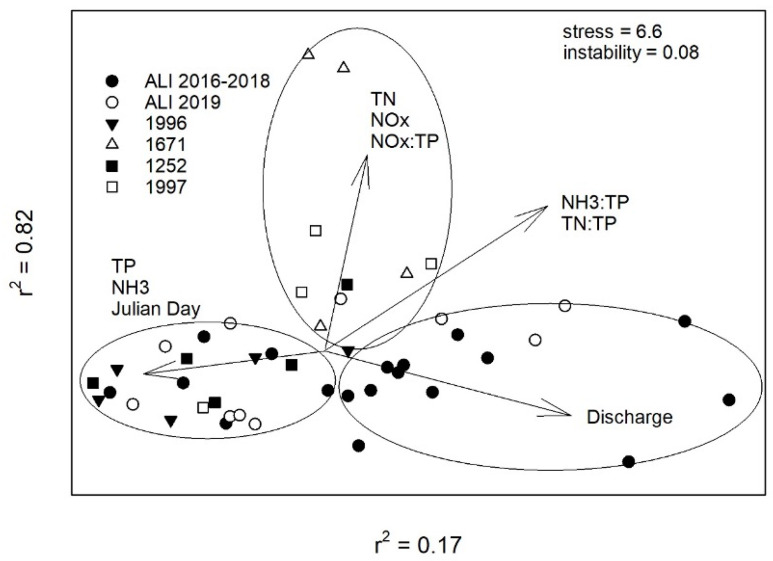
Non-metric multidimensional scaling ordination of water quality parameters collected between June and October 2019 from long-term Austin Lakes Index (ALI) and cyanoHAP monitoring sites. Ellipses are meant to help identify site groupings. Sites closer together have more similar environmental attributes than sites further apart. Correlation coefficients associated with each environmental variable are available in the corresponding dataset [26].

**Table 1 toxins-16-00091-t001:** Summary statistics of water quality parameters (mean ± standard deviation) collected from two sites (2 and 5) for the months of June–October of 2016–2018 (n = 18 except for NH_3_ where n = 17 and n = 10) as part of the Austin Lakes Index (ALI) routine monitoring program. Data were statistically compared with a student’s *T*-test or Mann–Whitney U-test when non-normality could be achieved. Additionally, summary statistics of nutrient concentrations and nutrient content from cyanoHAP monitoring sites (n = 8) are shown for comparison with results from the ALI sites. Parameters listed are ammonia nitrogen (NH_3_-N), nitrate+nitrite nitrogen (NO_x_-N), total Khjeldhal nitrogen (TKN), total nitrogen (TN), total phosphorus (TP), and the nitrogen to phosphorus molar ratio (N:P). Note: total suspended solids (TSS) were measured for sites 2 and 5 in 2016–2018 (3.3 ± 1.6 mg/mL) and 2019 (2.0 ± 0.8^+^ mg/mL), but TSS data were not available for the other sites.

Site(s)	NH_3_-N (µg/L)	NO_x_-N (µg/L)	TKN (µg/L)	TN (µg/L)	TP (µg/L)	N:P (Molar)
2 and 5 *	8.0 ± 0.01	199.0 ± 198.0	411.0 ± 82.9	610.0 ± 212.0	14.1 ± 9.3	123.4 ± 53.4
2 and 5 **	19.1 ± 18.2 ^+^	476.0 ± 488.0 ^++^	377.0 ± 49.9	853.0 ± 494.0	26.4 ± 15.5 ^+^	98.0 ± 70.6 ^+^
1996 ^#^	41.0 ± 7.7	112.5 ± 41.2	401.0 ± 28.7	513.5 ± 40.8	26.0 ± 20.8	87.1 ± 72.7
1671 ^#^	8.0 ± 0.0 ^ŧ^	1520.0 ± 176.9	184.7 ± 61.4	1704.7 ± 230.5	18.6 ± 9.3	259.4 ± 168.1
1252 ^#^	24.0 ± 20.3	393.8 ± 149.4	404.5 ± 56.5	798.3 ± 200.8	17.7 ± 6.5	119.1 ± 74.3
1997 ^#^	26.9 ± 32.7	390.7 ± 115.7	399.3 ± 16.6	790.0 ± 113.8	17.5 ± 16.4	165.6 ± 108.6

* ALI data for 2016–2018; ** ALI data for 2019; ^#^ cyanoHAP data for 2019; ^+^
*p* < 0.05; ^++^
*p* < 0.01; ^ŧ^ values were all below lab detection limits.

## Data Availability

Raw and processed data associated with this investigation are available through Mendeley Data [26].

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
