# Peer review of "Environmental Factors Impacting the Development of Toxic Cyanobacterial Proliferations in a Central Texas Reservoir"

_toxins, 2024, doi:10.3390/toxins16020091_

Round 1
Reviewer 1 Report
Comments and Suggestions for Authors
The manuscript is well written and the idea is clear. This work indeed will contribute to better understanding the emergence, duration, and potency of the cyanobacterial harmful algal blooms events. I would suggest to mention also the issue of cyanotoxins in the developing countries such as
Toxins 2021, 13(11), 786; https://doi.org/10.3390/toxins13110786 Figure 2 needs to be in a high resolution as the details are difficult to read.Author Response
Please see attachement.

Reviewer 2 Report
Comments and Suggestions for Authors
The MS reported the environmental factors impacting the development of toxic cyanobacterial proliferation a central Texas reservoir. This study achieved good results, providing reliable data for exploring the nature of the turning red in Lake Avernus. Generally, the MS contains valuable information for this topic, but there are still some issues need to improve.
1. English language still need improve.
2. subtitles need to change to contain more specific and complete information.
3. Line 124, 125, 127, 2019 strains,refer to the year of 2019? or the number of the strains? It cause misunderstanding of the meaning.
4. Figure 4 need to change to improve the resolution. It’s not clear enough.
5. Figure 5, what is the word, ave, for?
6. Table 1, add the explanation for the abbreviations such as TSS, et al. in the table note.
7. The letter p for statistical significance should be italic, check the whole MS.
8. Line 30, there is a syntax error, suggested to change "impacts to" to "impact on".
9. line 46, add "a" before "culture".
10. line 50, add "the" before "factors".
11. line 106, info, use the full term, information
12. line 113, delete "from" in the "sequenced from" sentence.
13. Line 129, the definite article "the" is missing before "identity".
14. Line 271, remove the "of" from "October of 2018".
15. Lines 278-283, suggest to restructure the sentences to make more engaging.
16. Line 292, the definite article "the" is missing before "the".
17. the phrase "due to the" appears many times in the text, and it is suggested to replace it with "because of".
18. line 258, add "the" before "cyanoHAB/HAP".
19. line 271, change "lead" to "leads".
20. line 332, change "are" to "is".
21. Please recheck thee references for format errors.
So, I suggested a minor revision to the MS.
Comments on the Quality of English LanguageEnglish language still need improve.
Reviewer 3 Report
Comments and Suggestions for Authors
The reviewed manuscript raises an important issue: the degradation of aquatic ecosystems, which leads not only to the deterioration of their ecological condition, but also to the threat to human and animal health caused by cyanobacterial toxins.
Although the manuscript can only be treated as a case study, the publication of the results will be an important comparative material for other studies.
It seems to require only minor changes.
First of all, the paragraph describing the purpose of the research should be rebuilt. Merely comparing the results (line 91) is an insufficient goal, and it is impossible to verify the hypothesis put forward in lines 93-94 due to lack of data. The description of the research hypothesis should directly refer to the statistical analyzes performed and the results obtained.
A few detailed notes:
The first sentence of the abstract is worded strangely. It can be understood that the death of animals is the main problem related to water blooms, which is not true, because blooms primarily cause many environmental and economic problems. This fragment should be changed.
Fig. 1 It seems that the information in the title "a part of the Austin Lake Index" should either be removed (this is explained in the methods) or should be expanded. For the non-Texas reader, it is unclear what the Austin Lake Index is.
Fig. 4 The diagram is illegible
Fig. 5. What was the concentration of toxins?
Reviewer 4 Report
Comments and Suggestions for Authors
Environmental factors impacting the development of toxic cyanobacterial proliferations in a central Texas Reservoir.
This manuscript uses monitoring efforts to document the presence of cyanotoxins, specifically anatoxin and its congeners in a reservoir following reported pet deaths. Overall, it was very well written, and I only have a few comments and suggestions for the authors.
General comment:
Please define abbreviations the first time they are used – normally this would be in the methods, but since the organization puts the methods last, consider defining abbreviations in the introduction and results. Some terms, from what I can see, are never defined:
- OTUs
- BLAST
-These are not common abbreviations used outside of the author's field.
Figure 3: Please provide taxa identification of the metaphyton in each panel.
Figure 4: Maybe I’m making assumptions- are the boxed groups specific cyanobacterial genera/tax – if so, please provide the taxa that go with each boxed group- if that is not the case, more detail is needed in the caption describing what is shown in the figure.
Table 1: Are NOx and NH3 concentrations reported based on the molecule or as the amount of Nitrogen present? If the latter both should be reported as NOx-N and NH3-N
Line 180: Change from significant to significantly
Line 297: ammonia is already defined as NH3- please use abbreviations consistently after each term has been defined.
Lines 300-306 It appears that NOx and NO3 are being used interchangeably- please pick one abbreviation and stick with it
Lines 300 – 306: It appears that phosphate and SRP are being used interchangeably- please use one or the other.
Lines 300 – 306 (and Table 1): I see there is discussion of SRP at the sites, but this is not shown in the table with the other nutrients is there a reason this data was omitted from the table?
Line 308/309: is 0.08 µg/L correct? Please double-check that this shouldn’t be mg/L.
Line 342: change “were” to “are” unless these sites are no longer popular recreational areas
